# TransFNN: A Novel Overtemperature Prediction Method for HVDC Converter Valves Based on an Improved Transformer and the F-NN Algorithm

**DOI:** 10.3390/s23084110

**Published:** 2023-04-19

**Authors:** Sihan Zhou, Liang Qin, Hui Sun, Bo Peng, Jiangjun Ruan, Jing Wang, Xu Tang, Xiaole Wang, Kaipei Liu

**Affiliations:** 1School of Electrical Engineering and Automation, Wuhan University, Wuhan 430072, China; 2Electric Power Research Institute of State Grid Anhui Electric Power Co., Ltd., Hefei 230601, China

**Keywords:** HVDC, converter valve, cooling system, time series prediction, Transformer, clustering algorithm

## Abstract

Appropriate cooling of the converter valve in a high-voltage direct current (HVDC) transmission system is highly significant for the safety, stability, and economical operation of a power grid. The proper adjustment of cooling measures is based on the accurate perception of the valve’s future overtemperature state, which is characterized by the valve’s cooling water temperature. However, very few previous studies have focused on this need, and the existing Transformer model, which excels in time-series predictions, cannot be directly applied to forecast the valve overtemperature state. In this study, we modified the Transformer and present a hybrid Transformer–FCM–NN (TransFNN) model to predict the future overtemperature state of the converter valve. The TransFNN model decouples the forecast process into two stages: (i) The modified Transformer is used to obtain the future values of the independent parameters; (ii) the relation between the valve cooling water temperature and the six independent operating parameters is fit, and the output of the Transformer is used to calculate the future values of the cooling water temperature. The results of the quantitative experiments showed that the proposed TransFNN model outperformed other models with which it was compared; with TransFNN being applied to predict the overtemperature state of the converter valves, the forecast accuracy was 91.81%, which was improved by 6.85% compared with that of the original Transformer model. Our work provides a novel approach to predicting the valve overtemperature state and acts as a data-driven tool for operation and maintenance personnel to use to adjust valve cooling measures punctually, effectively, and economically.

## 1. Introduction

With the increase in global power demand, the security and stability of power systems are more dependent on high-voltage direct current (HVDC) transmission systems [1], in which the converter valve is the core component. If the operating temperature of the converter valve is too high, its internal thyristors and other components will be damaged, thus affecting the normal operation of the converter station and even the regional power grid [2]. The converter valve is equipped with a cooling system. Among its many operating parameters, the cooling water temperature is an important criterion for determining whether the converter valve is in an overtemperature state [3], and different types of cooling systems have diverse overtemperature thresholds. If the temperature of the valve cooling water exceeds the corresponding threshold, the converter valve will operate in the overtemperature state and will be irreversibly damaged if stronger cooling measures are not taken [4]. 

At present, the operation and maintenance personnel of converter stations mainly judge a valve’s future overtemperature state through manual experience and adopt the following emergency cooling measures [5,6]: (i) increasing the cooling power of the water circulation system; (ii) putting the standby auxiliary cooling equipment into operation; (iii) increasing the water level of the cooling water tank; (iv) distributing the load in the area to other converter stations. However, due to the lack of quantitative perception of overtemperature, operation and maintenance personnel are unable to take timely and appropriate cooling measures, which leads to two negative consequences: On one hand, for the sake of safety, they tend to adopt cooling measures that are stronger than actually needed, but this will cause a waste of energy and water resources and impose dispensable pressure on the operation of the regional power grid [7]; on the other hand, manual experience is not reliable enough to address the occasional overtemperature of a converter valve, so the core equipment cannot be protected to the maximum extent. Therefore, research on methods for predicting the overtemperature state of an HVDC converter valve can improve the perception of operation and maintenance personnel of future overtemperature states, which is of great significance for resource conservation and the stable operation of power grids.

Currently, the research on the overtemperature state of HVDC converter valves is insufficient. Berg et al. [8] and Zhou et al. [9] simulated thermodynamic models of two specific types of converter valves by using 3D simulation and numerical calculation methods so as to obtain a novel valve layout for better heat dissipation efficiency, but converter valves of other types need to be re-simulated before analysis, which is not universally applicable. Liu et al. [10] studied the structure of an internal reactor and other components of a converter valve and proposed a method for reducing the heat generated during the operation of a converter valve by reducing the valve’s power loss, but this method is still incapable of improving the perception of operation and maintenance personnel of the overtemperature of converter valves. Some researchers proposed a new series structure for cooling systems [11], which can reduce power loss and make the temperature of a converter valve more easily perceived and controlled, but it has no guiding significance for a large number of converter stations that are already equipped with the original models of cooling systems. Ma et al. [12] studied the relation between the cooling water temperature of a valve and the DC load, but the approximated relation was vague, and the study lacked further research on the relation between the cooling water temperature and other operating parameters of the converter valve. To sum up, the studies mentioned above lacked adequate predictive research and complete operating parameter analyses for existing converter valves; therefore, they cannot enable operation and maintenance personnel to punctually and appropriately adjust valve cooling measures by analyzing future overtemperature states of converter valves. Therefore, it is necessary to study methods of time-series prediction for the cooling water temperature of converter valves.

In the field of time-series prediction, many AI methods have relatively excellent performance. They can avoid the difficulty of mastering physical mechanisms and excel in building implicit mappings between input and output values. Many algorithms have been developed for time-series prediction, e.g., the back-propagation neural network (BPNN) [13], random forest (RF) [14], support vector regression (SVR) [15], recurrent neural network (RNN) [16], long short-term memory (LSTM) [17], and gated recurrent unit (GRU) [18]. Generally speaking, more recent algorithms have certain improvements in prediction accuracy and training complexity compared with earlier ones. The attention mechanism [19] is the basis of the novel Transformer encoder–decoder architecture [20], which was originally applied in the field of natural language processing (NLP). This Transformer model has powerful sequence–sequence prediction abilities; it can establish the global dependence between the input and output, and it has an excellent prediction effect on time series [21].

To the best of our knowledge, the Transformer model has different performance when processing time series in different scenarios [21]. Under the application background of this study, the valve cooling water temperature is not a typical time series, as it is influenced by multi-source parameters under different operating conditions. Therefore, we need to modify the Transformer model in order to equip it with better competence in valve overtemperature prediction. In this context, we propose a novel TransFNN model that coordinates an improved Transformer model and the F-NN framework (composed of the FCM algorithm and artificial neural networks). On one hand, the Transformer model can forecast future values of the independent parameters of a converter valve. On the other hand, F-NN can accurately fit the relations between several independent parameters and one dependent parameter of the converter valve, and it can calculate the future values of the valve’s cooling water temperature, i.e., the dependent parameter, by using future values of the independent parameters obtained by the Transformer. The major contributions and innovations of this study can be stated from three perspectives:(i)The proposed TransFNN model is capable of more accurately forecasting the overtemperature state of a converter valve, and it can act as a data-driven tool for operation and maintenance personnel to use to effectively and economically adjust cooling measures.(ii)We improved the positional encoding method in the Transformer to make it suitable for predictions of time series with strong periodicity in different time dimensions.(iii)Our research provides an effective framework for time-series predictions with multi-source data, and it has satisfying performance in scenarios in which the investigated object has a variety of complex operating conditions.

The remainder of this paper is organized as follows. Section 2 presents the mathematical model for the prediction of a valve’s cooling water temperature. In Section 3, the Transformer-based model is modified to address the problem of independent operating parameter prediction. Section 4 is devoted to introducing the method for the calculation of a valve’s cooling water temperature based on the F-NN algorithm. Section 5 involves quantitative experiments, as well as an evaluation of the result and an analysis. Section 6 summarizes the crucial conclusions and declares the research prospects.

## 2. Mathematical Model for Valve Cooling Water Temperature Predictions

The prediction of a valve’s cooling water temperature is essentially a combination of time-series prediction and function approximation, and the main stages of model construction are as follows: Firstly, the influencing factors of a valve’s cooling water temperature are identified as the independent parameters; secondly, a time-series prediction model is established to forecast future values of the independent variables; thirdly, the relation between the valve’s cooling water temperature and the independent variables is approximated by using the historical operating data of the converter valve; finally, the two models established in the second and third stages are combined to form the complete model. The model is trained with historical operating data so that it can be used to predict the future values of the valve’s cooling water temperature. A flowchart of the overall scheme is shown in Figure 1.

A valve cooling system can be divided into its internal and external parts. The internal cooling water system is a closed cycle and has the functions of heat absorption, water circulation, water treatment, etc. The cooling water in the internal system absorbs the heat generated during the operation of the converter valve and is sent to the external cooling water system. The external cooling water system is an open cycle with the functions of water softening, water purification, water make-up, etc. A cooling tower sprays the internal cooling water pipe for heat dissipation, and the heat of the internal cooling water system is discharged into the atmospheric environment through a fan. The internal and external cooling water systems work together to perform continuous cooling for the converter valve.

Generally, when the operation and maintenance personnel of a converter station patrol the cooling system of a converter valve, eight operating parameters are recorded, and they can be divided into three categories:(i)Electrical data: DC load;(ii)Environmental data: external ambient temperature, valve cooling water temperature;(iii)Water cooling system data: inlet water temperature of the valve’s cooling system, outlet water temperature of the valve’s cooling system.

Both the inlet and outlet water temperatures can be regarded as parameters of the valve’s cooling water temperature, which indicates whether the converter valve is overheated. The inlet water temperature and the cooling measures are two main factors that determine the outlet water temperature. Since the cooling measures of a converter valve are not the main focus of this study and the inlet and outlet water temperatures show a high correlation (the correlation analysis is presented in Section 5), we ignore the parameter of outlet water temperature and use the inlet water temperature to represent the valve’s cooling water temperature, i.e., the dependent parameter. The other six operating parameters are defined as independent variables, and they have a decisive impact on the dependent variable, as shown in Table 1.

For parameter predictions, we can take the DC load (P) as an example. We denote the historical sequence of P at time t as $x_{t}$, and our target is to predict the future P time series $y_{t}=\left(y_{t+1},y_{t+2},\cdots ,y_{t+q}\right)$ at time t, where q represents the maximum number of prediction steps and $y_{t+1}$ is the value of the DC load at time $\left(t+1\right)$. The prediction model for the time series is expressed by Equation (1):(1)$$y_{t}=f\left(x_{t}|\theta \right)$$
where θ represents model’s parameter vector; $f\left(\cdot \right)$ is the implicit function of the time-series prediction model to be introduced in Section 3; $x_{t}$ represents the input historical sequence, as shown in Equation (2):(2)$$x_{t}=\left(x_{t-n+1},x_{t-n+2},\cdots ,x_{t}\right)\in R^{n}$$
where n and $x_{t}$ represent the length of the historical sequence and the value of the DC load at time t, respectively.

The prediction model is trained on the historical dataset. In a given dataset consisting of N input–output pairs $\left\{\left(x_{1},y_{1}\right),\left(x_{2},y_{2}\right),\cdots ,\left(x_{N},y_{N}\right)\right\}$, the optimal parameter vector $\hat{\theta }$ of the trained model is obtained by minimizing the loss function as follows:(3)$$loss=\frac{1}{N}\sum_{t=1}^{N}\left\|y_{t}-{\hat{y}}_{t}\right\|$$
(4)$${\hat{y}}_{t}=f\left(x_{t}|\hat{\theta }\right)$$

In addition to the DC power, the above model can also be used to predict the future values of the external ambient temperature and the valve hall temperature. However, some independent parameters, i.e., the water level of the expansion tank, the water level of the spray tank, and the conductivity of the main water circuit, are directly controlled by the operation and maintenance personnel [5], so they do not have typical temporal characteristics. Therefore, when predicting the future values of these three parameters, their future values that were preset by the operation and maintenance personnel are generally accepted, and the other three with typical time-series characteristics can be predicted by using the improved Transformer model, which will be introduced in Section 3.

For this multi-input and single-output prediction problem, we also need a model to approximate the relation between the several inputs and the one output. The value of the dependent variable at time t is denoted as $o_{t}$, and the vector of independent variables is denoted as $i_{t}$, which includes g different types of features. Then, the implicit function between the independent and dependent variables can be represented by $F\left(\cdot \right)$, as shown in Equation (5):(5)$$o_{t}=F\left(i_{t}\right)=F\left[\left(i_{t}^{\left(1\right)},i_{t}^{\left(2\right)},\cdots ,i_{t}^{\left(g\right)}\right)\right]$$

All independent and dependent parameters are substituted into Equation (5):(6)$$T_{t}=F\left[\left(P_{t},T_{t}^{S},T_{t}^{P},L_{t}^{P},L_{t}^{S},\sigma_{t}\right)\right]$$
where g=6, indicating that there are 6 independent variables; $F\left(\cdot \right)$ is the implicit function representing the relation between the 6 independent variables and the dependent variable, which is to be introduced in Section 4.

It is worth mentioning that the simple time-series prediction of a valve’s cooling water temperature is of little reference value. It can be seen in Equation (6) that the valve’s cooling water temperature is determined by the six independent variables, but only P, $T^{S},$ and $T^{P}$ are typical time-series parameters, while $L^{P}$,$L^{S},$ and σ have few temporal characteristics. Therefore, if the valve’s cooling water temperature is directly predicted in time series, it is the default that all the corresponding independent variables have typical time-series characteristics. This will lead to relatively large errors in the prediction results because this forecasting method runs contrary to the temporal characteristics of the operating parameters; a detailed verification of this will be carried out in Section 5. Hence, we adopt the following prediction strategies:P, $T^{S},$ and $T^{P}$, which have obvious time-series characteristics, will participate in the time-series prediction, while $L^{P}$, $L^{S},$ and σ, which are directly controlled by the operation and maintenance personnel, will be preset before the forecasting process; when the values of the independent variables at time t are given, the predicted value of the corresponding valve cooling water temperature can be calculated by using Equation (6). The process of predicting a valve’s cooling water temperature introduced in this section is demonstrated in Figure 2.

## 3. Time-Series Prediction Model Based on the Improved Transformer

The encoder–decoder architecture is based on the attention mechanism, and it was originally used in the field of NLP. The Transformer is a more novel model based on the encoder–decoder architecture, and it has great application prospects in the domain of time-series prediction. In this section, the attention mechanism, the modified Transformer model, and the process of time-series prediction by the Transformer model will be introduced.

### 3.1. Fundamentals of the Attention Mechanism

The attention mechanism fundamentally maps a query to a series of key–value pairs, which are regarded as the constituent elements within a source. There are three steps in computing attention [20]: Firstly, the similarity of the *Query* and *Key* is calculated by using a perceptron, dot product, or concat operation; secondly, the similarity calculated above is normalized by a softmax function; finally, the sum of the weights multiplied by the corresponding *Value* is calculated so that the attention value can be obtained. The three steps can be summarized as follows:(7)$$Atten\left(Query,Source\right)=\sum_{i=1}^{L_{x}}Similarity\left(Query,Key_{i}\right)\cdot Value_{i}$$
where $L_{x}$ represents the length of the source.

The scaled dot-product attention model [20], as shown in Figure 3a, provides a more practical attention calculation method. Its input contains a $d_{k}$-dimensional Query, $d_{k}$-dimensional Key, and $d_{v}$-dimensional Value, and its output is
(8)$$Atten\left(Q,K,V\right)=\mathrm{softmax}\left(QK^{T}/\sqrt{d_{k}}\right)\cdot V$$
where $\sqrt{d_{k}}$ is used to minimize the inner product; $Q\in R^{n\times d_{k}}$, $K\in R^{m\times d_{k}},$ and $V\in R^{m\times d_{v}}$ represent the Query, Key, and Value, respectively; $QK^{T}$ is used to evaluate the similarity between the Query and Key. In the attention mechanism, K is generally identical to V, and the normalized similarity can be obtained after $QK^{T}$ is processed with a softmax function. In the scaled dot-product attention mechanism, Q, K, and V are calculated as follows: $Q=W_{q}X$, $K=W_{k}X$**,** and $V=W_{v}X$, where X is the input matrix of the attention calculation, and $W_{q}$, $W_{k},$ and $W_{v}$ are the weight matrices, which are randomly initialized and updated during the training of the model, so the values of the output of Equation (8) can better reflect the importance of each element within the sequence.

The multi-head attention model [20] is an improvement of the scaled dot-product attention model, as shown in Figure 3b. In this model, Q, K, and V are mapped through the projection matrices, after which they are used to perform the attention calculation. Then, there are r repetitive processes, whose results are finally concatenated, as shown in Equation (9):(9)$${head}_{p}=Atten(QW_{p}^{Q},KW_{p}^{K},VW_{p}^{V})(p=1,2,\cdots ,r)$$
where $W_{p}^{Q}\in R^{d_{model}\times d_{k}}$,$W_{p}^{K}\in R^{d_{model}\times d_{k}},$ and $W_{p}^{V}\in R^{d_{model}\times d_{v}}$ are projection matrices that project Q, K, and V into r different dimensions, so different attention models can be trained, leading to an improvement in the accuracy of the attention calculation. Then, the value of attention of r heads is concatenated into one, as shown in Equation (10):(10)$$Multihead(Q,K,V)=concat({head}_{1},{head}_{2},\cdots ,{head}_{r})$$
where $W^{O}\in R^{{rd}_{v}\times d_{model}}$. 

### 3.2. The Transformer Model

The Transformer model is a novel architecture based on the attention mechanism, and it excels in processing sequence-to-sequence problems. It is composed of an encoder and a decoder, and the architecture is shown in Figure 4.

The encoder consists of an input layer, a positional encoding layer, and several stacked encoder layers that share the same structure and parameters. Each encoder layer contains a multi-head attention layer and a feed-forward layer, both of which are followed by a normalization layer. The input sequence is mapped to be $d_{model}$-dimensional and coded to get the order information in the positional encoding (PE) layer. The PE method of our proposed model is different from the absolute positional encoding method adopted in the original Transformer model [20]. The DC load, external ambient temperature, and valve hall temperature are the three independent parameters to be predicted, and they all have obvious periodicity in the fluctuations in their values over 24 h per day, 30 days per month, and 12 months per year. Therefore, it is necessary to separately perform positional encoding in the three dimensions, and an example of hourly positional encoding is shown in Equations (11) and (12).
(11)$$PE_{\sin}^{1}\left(hour\right)=\sin\left(2\pi \cdot \frac{hour}{\max\left(hour\right)}\right)$$
(12)$$PE_{\cos}^{1}\left(hour\right)=\cos\left(2\pi \cdot \frac{hour}{\max\left(hour\right)}\right)$$
where ${PE}_{sin}^{1}\left(\cdot \right)$ and ${PE}_{cos}^{1}\left(\cdot \right)$ represent the sine and cosine forms of PE in the first dimension (hourly) of a certain element from the input sequence, respectively; hour represents the hour of the day corresponding to the time when this element was sampled, and $max\left(hour\right)=24$. When positional encoding needs to be performed daily or monthly, $max\left(day\right)$ and $max\left(month\right)$ are 30 and 12, respectively. Thus, the input sequence is mapped into 7 dimensions, with the original value in the first dimension, and ${PE}_{sin}^{1}\left(\cdot \right)$, ${PE}_{cos}^{1}\left(\cdot \right)$, ${PE}_{sin}^{2}\left(\cdot \right)$, ${PE}_{cos}^{2}\left(\cdot \right)$, ${PE}_{sin}^{3}\left(\cdot \right)$, and ${PE}_{cos}^{3}\left(\cdot \right)$ are the elements of the other six dimensions. By improving the PE, the periodicity of the independent parameters of the converter valve in different time dimensions can be better learned by the Transformer model. 

The output sequence of the encoder contains abundant information, and it is fed into the decoder for further processing. In order to use the encoder’s output for the attention mechanism, the decoder layer inserts an encoder–decoder multi-head attention layer between the multi-head layer and the feed-forward layer. After each sublayer in the decoder, there is also a normalization layer, and the linear mapping layer processes the output of the last decoder layer to obtain the decoder’s output.

In Figure 4, the input of the encoder is a sequence $x_{t}=\left(x_{t-n+1},x_{t-n+2},\cdots ,x_{t}\right)$, i.e., historical DC load data, where t and n represent the current time step and the length of the input sequence, respectively. The decoder’s input is $y_{t-1}=\left(y_{t},y_{t+1},\cdots ,y_{t+q-1}\right)$, i.e., future DC load data, where q is the maximum prediction step. The decoder’s output is $y_{t}=\left(y_{t+1},y_{t+2},\cdots ,y_{t+q}\right)$, which lags the input sequence of the decoder by one step. The parameters of the Transformer model are obtained through training, e.g., with the weights and biases of the feed-forward layers and the values of $W_{q}$, $W_{k},$ and $W_{v}$ in the attention layers.

### 3.3. Process of Parameter Prediction Based on the Transformer

The process of independent parameter predictions with the Transformer model is very similar to that of other time-series prediction models, such as LSTM [22]. The overall prediction process is shown in Figure 5, and all variables to be predicted, including the DC load, ambient temperature, and valve hall temperature, need to go through the same process, but by using three trained Transformer models with different parameter vectors $\hat{\theta }$.

Taking the prediction of the DC load as an example, the steps are as follows:
(i)The historical DC load data are collected as the input, and they are mapped to [0,1]. The normalization formula is in shown in (13):(13)$$x\to x^{N}=\frac{x-x_{\min}}{x_{\max}-x_{\min}}$$
where x is the original sample data, and $x^{N}$ is the normalized form; $x_{max}$ and $x_{min}$ represent the maximum and minimum values of the sample dataset, respectively.(ii)The sample dataset is divided into a training set, validation set, and test set.(iii)Candidate Transformer models with different hyperparameters are obtained by using manual experience and grid search methods [23].(iv)The candidate models are trained on the training set.(v)The trained candidate models are evaluated on the validation set and the optimal Transformer model that produces the lowest error is selected.(vi)The optimal Transformer model is tested on the test set. The output of the optimal Transformer model is denormalized to obtain the predictive values, and they are analyzed with some evaluation indicators [24].

## 4. Calculation of a Valve’s Cooling Water Temperature Based on F-NN

In Equation (6), the implicit function $F\left(\cdot \right)$ represents the relation between the valve’s cooling water temperature and the six independent parameters. These seven variables involve complex thermodynamic and electromagnetic relations; therefore, it is difficult to obtain analytical expressions. However, deep learning methods can be used for function approximation so as to fit the relation between the valve’s cooling water temperature and the six independent variables. A converter station operates in different seasons and time periods, and it may be in different operating conditions due to the influence of the external environment and the peaks and valleys of power consumption [1]. Therefore, we put forward the F-NN algorithm, which coordinates the fuzzy C-means clustering (FCM) algorithm and artificial neural networks (ANNs). Firstly, the FCM algorithm is used to cluster the data of the valve’s operating parameters and to form several typical databases belonging to different typical operating conditions of the converter valve. Then, a neural network is applied to fit the relationship between the independent variables and dependent variables under different operating conditions obtained by FCM. By using F-NN, we can approximate the relation represented by the implicit function $F\left(\cdot \right)$ and then realize the lean calculation of the cooling water temperature of the converter valve.

### 4.1. Extraction of Typical Operating Conditions Based on FCM

#### 4.1.1. Fundamentals of the FCM Algorithm

FCM is a partition-based clustering algorithm that ensures that the samples clustered in the same class have the highest similarity and that the samples between different classes have the lowest similarity [25]. The FCM algorithm introduces the concept of the membership degree, making it distinct from the K-means clustering algorithm. The membership degree matrix is used to judge the degree of belonging of each sample to each category. The probability of samples belonging to each category is between 0 and 1, so the FCM clustering algorithm is deemed to be soft clustering. The core idea of the FCM algorithm is to constantly update the membership matrix and clustering center matrix until the objective function reaches the minimum, and then the clustering center of the final iteration will be output.

The original operating database of the converter valve can be divided into an input matrix and an output sequence. We denote the historical data matrix of independent parameters in the valve cooling system as the input matrix I, as shown in Equation (14):(14)I=(PTSΤPLPLSσ)=(i(1)i(2)⋯i(6))∈ℝN×6
where ik=i1k,i2k,⋯,ijk,⋯,iNkT∈RN×1 is the input sequence of the k type of independent parameter, and N is the length of the historical data sequence. The diagram of the original operating database of the converter valve is shown in Figure 6.

The objective function of the FCM algorithm is:(15)$$\left\{ \begin{array}{l} \min J\left(U,V\right)=\sum_{h=1}^{c}\sum_{j=1}^{N}{\left(\mu_{hj}\right)}^{m}{\left(d_{hj}\right)}^{2}\\ s.t.\mu_{hj}\in \left[0,1\right],\forall h,j;\sum_{h=1}^{c}\mu_{hj}=1,\forall j \end{array}\right.$$
where c represents the total clustering categories that should be set in advance, and it has a great impact on the validity of clustering; h=1,2,⋯,c is the category order number; $U={\left[\mu_{hj}\right]}_{c\times N}$ is the membership matrix, in which all elements in a column add up to 1, and every element $\mu_{hj}\in \left[0,1\right]$ stands for the membership of the j sample in the h category; $V={\left[v_{hk}\right]}_{c\times g}$ is the clustering center matrix, in which any element $v_{hk}$ stands for the clustering center of the h category for the k variable feature; $m\in \left[1,+\infty \right]$ is the fuzzy index, and the clustering effect is best when m=2 [26]; $d_{hj}=\left\|i_{j}-v_{h}\right\|$ is the Euclidean distance of the input vector of the j sample and the h clustering center. Both U and V are randomly initialized.

Lagrange multiplication is used to optimize the objective function to obtain the following formula:(16)$$J=\sum_{h=1}^{c}\sum_{j=1}^{N}{\left(\mu_{hj}\right)}^{m}{\left(d_{hj}\right)}^{2}+\lambda \left(\sum_{h=1}^{c}\mu_{hj}-1\right)$$

The necessary first-order conditions for optimization are expressed as follows:(17)$$\frac{\partial J}{\partial \lambda }=\left(\sum_{h=1}^{c}\mu_{hj}-1\right)=0$$
(18)$$\frac{\partial J}{\partial \mu_{hj}}=\left[m{\left(\mu_{hj}\right)}^{m-1}{\left(d_{hj}\right)}^{2}+\lambda \right]=0$$
(19)$$\frac{\partial J}{\partial d_{hj}}=2{\left(\mu_{hj}\right)}^{m}\left(i_{j}-v_{h}\right)=0$$

According to Equations (17)–(19), the relationship between the membership and Euclidean distance of each element can be obtained, as shown in Equation (20).
(20)$$\mu_{hj}=d_{hj}{}^{\frac{2}{1-m}}/\sum_{h=1}^{c}{\left(d_{hj}\right)}^{\frac{m-1}{2}}$$

Hence, the clustering center can be derived from (19) and (20),
(21)$$v_{h}={\sum_{j=1}^{N}\left(\mu_{hj}\right)}^{m}i_{j}/{\sum_{j=1}^{N}\left(\mu_{hj}\right)}^{m}$$

Thus, by Equations (20) and (21), the clustering center matrix and its corresponding membership matrix can be obtained. l iterations are performed to update U and V with Equations (20) and (21) until the value change of the objective function is smaller than the preset iteration accuracy $\varepsilon \left(\varepsilon >0\right)$, as shown in (22):(22)$$J\left[U^{\left(l\right)},V^{\left(l\right)}\right]-J\left[U^{\left(l-1\right)},V^{\left(l-1\right)}\right]<\varepsilon$$
where $U^{\left(l\right)}$ and $V^{\left(l\right)}$ represent the membership matrix and the clustering center matrix in the $l^{th}$ iteration, respectively.

FCM is an unsupervised algorithm, and it is necessary to preset a specific clustering category number c so that the sample data will be clustered into c categories. Based on the fundamental principle of the clustering algorithm, the smaller the overall distance within a cluster is, and the larger the distances between clusters are, and the more valid the clustering result will be. The clustering category number c has a decisive influence on the validity of the clustering results; therefore, a function of c is required to characterize the clustering validity, which is named the clustering validity function [26], as shown in Equation (23):(23)$$P\left(c,U,V\right)=\frac{\min_{h=1}^{c}\left(\sum_{j=1}^{N}\mu_{hj}\right)}{\max_{h=1}^{c}\left(\sum_{j=1}^{N}\mu_{hj}\right)}\left[\frac{1}{c}\sum_{h=1}^{c}\left(\frac{\sum_{j=1}^{N}\mu_{hj}^{2}}{\sum_{j=1}^{N}\mu_{hj}}\right)+1-\frac{\sum_{h=1}^{c}\sum_{j=1}^{N}\mu_{hj}^{2}{\left\|i_{j}-v_{h}\right\|}^{2}}{\sum_{j=1}^{N}{\left\|i_{j}-\left(\sum_{j=1}^{N}i_{j}\right)/N\right\|}^{2}}\right]$$

The greater the value of $P\left(c,U,V\right)$ is, the more effective the clustering results are. Before determining the clustering category number, we can let c traverse $c_{max}$ integers: $c\in \left\{1,2,\cdots ,c_{max}\right\}$. Then, the FCM calculation process is performed based on different values of c so as to obtain the optimal $c^{*}$ corresponding to the maximum value of $P\left(c,U,V\right)$, as shown in Equation (24):(24)$$c^{*}=\arg\max_{c}\left[P\left(c,U,V\right)\right]$$

The pseudocode of the FCM algorithm is illustrated in Algorithm 1.
**Algorithm 1:** Pseudocode of the FCM algorithmSet the fuzzy index m, iteration accuracy ε, and maximum category number cmax**for** c=1,2,⋯,cmax    l=0
    Initialize Vl∈Rc×g and Ul∈Rc×N randomly    **while** True         l=l+1
        Update Ul with Equation (20)        Update Vl with Equation (21)        **if** [In Equation (22) is satisfied]            **break while**        **end if**    **end while**    U=Ul,V=Vl
    Calculate the clustering validity function Pc,U,V with Equation (23)**end for**Choose the optimal c(c*) with Equation (24)return c* and the corresponding V,U


#### 4.1.2. Construction of Typical Operating Databases

By using the optimal c that was obtained, *N* samples of historical operating data can be clustered into c categories corresponding to c typical databases, each of which contains $N_{1},N_{2},\cdots ,N_{c}$ samples consisting of six independent variables and one dependent variable from Table 1. The h typical database belongs to the h typical operating condition, which is mathematically characterized by the clustering center vector $v_{h}$. The data vectors in a typical operating database have the highest similarity with each other and the greatest distinction from those in other typical operating databases. A diagram of the construction of a typical database is shown in Figure 7.

### 4.2. Relation Approximation on Different Operating Conditions Based on an ANN

For each database representing a typical operating condition, we need to fit the relation between the independent parameters and the dependent parameter so that when the input vector is given under a certain operating condition, we can accurately calculate its corresponding output value, i.e., the valve’s cooling water temperature. In this context, of the many types of artificial neural networks, the back-propagation neural network (BPNN) algorithm is of high applicability.

Since each operating condition indicates a distinct relation between the dependent and independent parameters, c neural networks should be separately trained to approximate the relations. They share the same network structure and hyperparameters, e.g., the number of hidden layers and the number of neurons in each hidden layer, and they all have 6 inputs and 1 output, but the weight and bias parameters of each neural network are different. When the hyperparameters and network parameters of c neural networks are determined, we can use the FCM-based division of typical operating conditions and the ANN-based relation approximation of different operating conditions to calculate the valve’s cooling water temperature if a set of independent parameters are given.

When we need to calculate the value of a valve’s cooling water temperature $T_{j}=o_{j}$ corresponding to a new input vector $i_{j}$, we should first determine the type of typical operating condition to which $i_{j}$ belongs by calculating the optimal $h\left(h^{*}\right)$. The Euclidean distance between $i_{j}$ and the $h^{*}$ clustering center $v_{h}$ is minimized, as shown in Equation (25):(25)$$h^{*}=\arg\min_{h}\left(\left\|i_{j}-v_{h}\right\|\right)$$

$i_{j}$ is fed into an NN trained on the $h^{*}$ typical database, and the output of the NN is the valve’s cooling water temperature that is to be predicted. So far, the implicit function $F\left(\cdot \right)$ in Equation (6) has been realized with F-NN, and we can use TransFNN, which coordinates the improved Transformer model and F-NN, to predict future values of the valve’s cooling water temperature.

## 5. Experiments

In this section, we use the proposed TransFNN model to predict a valve’s cooling water temperature. The dataset division, correlation analysis, model hyperparameter settings, quantitative experiments, and result analysis are involved.

### 5.1. Dataset Division

This experiment used the 2012–2014 operating data from 12 converter valves of three HVDC converter stations in different regions of China. The historical operating data included eight types of feature quantities, which were introduced in Section 2, and the sampling interval of the data was 4 h. The 12 converter valves were denoted as No. 1–12. For each converter valve, the dataset produced by the operation and maintenance personnel contained 6576 records (from 1 January 2012 at 00:00 a.m. to 31 December 2014 at 20:00 p.m.). In the case of failure or maintenance, the converter valve was shut down, so the data of each converter valve within 3 years had several breakpoints in time, and the data of 3 years were, thereby, cut into several continuous data sequences.

There were 12 prediction tasks corresponding to 12 converter valves. Taking the time breakpoints of the dataset of each converter valve as the dividing points, each dataset was divided into a training set, a validation set, and a test set, with a ratio of approximately 3:1:1. All training sets were used for model training, the validation sets were devoted to selecting the candidate models with the optimal hyperparameters, and the test sets were used to evaluate the prediction results.

### 5.2. Correlation Analysis of Independent Variables

In Section 2, we introduced eight operating parameters that are generally recorded by the operation and maintenance personnel of converter stations, i.e., the DC load, external ambient temperature, valve hall temperature, water level of the expansion tank, water level of the spray tank, main water circuit conductivity, valve inlet water temperature, and valve outlet water temperature. However, there can be some correlations between one parameter and another. If there are parameters that are highly correlated, the complexity of the model will be pointlessly magnified, and the training speed of the model will, thus, be reduced. Therefore, we used the Pearson correlation coefficient [27] to perform a correlation analysis of the operating parameters, and the parameters with high correlation coefficients with others were eliminated.

The results of Pearson’s correlation analysis are shown in Figure 8. It was apparent that the inlet water temperature and the outlet water temperature were highly correlated, with the correlation coefficient being as high as 0.96. Therefore, the output water temperature of the converter valve was eliminated from the list of operating parameters that we studied, and the inlet water temperature was used to represent the valve’s cooling water temperature. In addition, all independent parameters appeared competent in acting as the input variables of the proposed prediction method; on one hand, the Pearson correlation coefficient between any two of the independent parameters was less than 0.5 or greater than −0.5, which meant that all six independent parameters mentioned in Table 1 were not highly relevant; on the other hand, each of the independent parameters showed a strong link with the inlet water temperature, as their Pearson correlation coefficients were all greater than 0.5 or smaller than −0.5. Therefore, the six independent parameters were used to predict the valve’s cooling water temperature.

### 5.3. Model Hyperparameter Settings

In the proposed TransFNN model, there are three types of hyperparameters to be set: (i) the clustering category number c of the FCM algorithm, (ii) the length of the historical data sequence n and the maximum prediction step q, (iii) and the number of hidden layers, the number of neurons in each hidden layer of the BPNN, and the structural and learning parameters of the improved Transformer model.

We first used the 12 training sets of the 12 converter valves to calculate the value of the clustering validity function according to Equation (25). We let the category number c traverse from 1 to 10, and the results for each converter valve are shown in Figure 9. For valves No. 6 and No. 12, the values of the clustering validity function corresponding to c=4 were marginally greater than those for c=3, and for the other converter valves, the validity of the clustering results was the highest when c=3. Therefore, the optimal clustering number c was set to 3 for this case.

The length of the historical data sequence n is of great significance in the Transformer model. To determine n, we should take both the completeness of the temporal information and the training complexity into consideration. If n is determined to be too large, the training time will be exceedingly long; if it is too small, the input data sequence will not contain sufficient temporal information, which will cause a reduction in prediction accuracy. Therefore, the principle of determining n is that the training complexity should be reduced as much as possible on the premise of ensuring the sufficiency of temporal information in the input historical data sequence. Trappenberg et al. [28] proposed a method for calculating the correlation of the mutual information between $x\left(t\right)$ and $x\left(t+\delta \right)$, where $x\left(t\right)$ and $x\left(t+\delta \right)$ represent a time series and its corresponding delay sequence, respectively, and δ is the length of the lag. In the case that we studied, the mutual information correlation (MIC) between $x\left(t\right)$ and $x\left(t+\delta \right)$ showed a downward trend overall, indicating that the two sequences contained more different temporal information as δ increased. When δ was between 40 and 112, the value of MIC was almost constant. For the Transformer, the value of n was determined by the preset threshold of MIC. When MIC was set to approximately 8, the corresponding n was between 40 and 112, and the setting of the MIC threshold was based on manual experience [23]. Given that $n\in \left[40,112\right]$ will cause the input historical data sequence to have the same degree of temporal information sufficiency, and considering the training complexity, n should be as small as possible, so we set n to 40. In addition, the larger q is, the longer the overtemperature state of the converter valve can be monitored, but the predicted values have decreasing reference value as the time step increases. In practical engineering applications, 2 days are adequate for the operation and maintenance personnel of a converter station to adjust the future cooling measures, and there were six samples within a day in this case, so we set q to 12.

In the BPNN and the Transformer model, there are eight main hyperparameters to be determined. Using manual experience and grid search methods, the eight structural and learning parameters were set as follows: (i) The number of hidden layers in the BPNN was 2; (ii) the numbers of neurons in each hidden layer were 10 and 5, respectively; (iii) the number of encoding/decoding layers was 3; (iv) the number of neurons in each hidden layer of the feed-forward layer was 32; (v) the learning rate was 0.01; (vi) the size of the epochs was 32; (vii) the batch size was 16; (viii) the dropout was 0.5.

### 5.4. Quantitative Analysis of TransFNN: Evaluating the Impacts of Individual Modules

In this experiment, historical data sequences of 40 time steps were used to predict future sequences of 12 time steps. It should be noted that DC power, external ambient temperature, and valve hall temperature have obvious time-series characteristics, so their future values were given by the output of the Transformer model; the water level of the expansion tank, water level of the spray tank, and main water circuit conductivity were directly controlled by the operation and maintenance personnel, so their future values in the experiment were obtained by adding Gaussian white noise with an average of 0 to the actual values, where the Gaussian white noise was used to simulate the error of reading the indications of the equipment.

The quantitative experiments were performed with the following models:
(i)The proposed TransFNN; (ii)TransFNN−FCM: the TransFNN model without the FCM module; (iii)TransFNN−ANN: the TransFNN model in which the ANN module was substituted with a simpler one, e.g., a multiple linear regression model; (iv)TransFNN−PE: the TransFNN model without the improved positional encoding method; (v)The Transformer model only: equivalent to the TransFNN model without the FCM and ANN modules. The valve’s cooling water temperature was directly predicted with the Transformer.

Each converter valve required a single prediction task. We evaluated the five models on all test sets of every converter valve with statistical measures, i.e., RMSE, MAE, sMAPE, and $R^{2}$. The results are shown in Table 2, which indicates that the rankings in every prediction task were not identical, but the TransFNN model always performed the best among the five models in the 12 prediction tasks.

For the sake of clarity, converter valve No. 7 was chosen to display the illustrative results, as shown in Figure 10, and we also present a scatterplot of the measured and predicted values of the valves’ cooling water temperatures in Figure 11.

It was indicated after comparing the deviation between the actual values and the predicted values that the proposed TransFNN was the best model for predicting the future values of valves’ cooling water temperatures among the five models. The reasons for the reductions in prediction accuracy of the other four models were as follows:
(i)TransFNN−FCM: A single neural network needed to process the data of all operating conditions because the valve operating conditions were not clustered into several typical operating conditions due to the absence of the FCM module. The learning and fitting abilities of the single NN were insufficient.(ii)TransFNN−ANN: The complex nonlinear relation between the six independent variable parameters and the valves’ cooling water temperatures was too complicated for simpler models to approximate.(iii)TransFNN−PE: Without the improved positional encoding method, the model was incapable of effectively learning the periodicity of the predicted variables in the dimensions of hours, days, and months.(iv)Only Transformer: Not all impact factors of the valves’ cooling water temperature had temporal characteristics because some factors were directly controlled by the valve operation and maintenance personnel. Therefore, the data sequence of valves’ cooling water temperatures cannot be regarded as a fully typical time series; otherwise, there would be a relatively large error in the predicted values. This also confirmed the discussion of the temporal characteristics of valves’ cooling water temperatures in Section 2.

A converter valve will be in an overtemperature state if the valve’s cooling water temperature exceeds a threshold. In order to evaluate the accuracy of the five models in predicting the overtemperature state of a converter valve, we preset the threshold of the cooling water temperature to 38 °C. When the converter valve was actually in the overtemperature state but the cooling water temperature of the valve did not exceed 38 °C or when the converter valve was not in the overtemperature state but the predicted value indicated an overtemperature state, the predicted result was considered to be wrong. We counted the error rate of overtemperature prediction of each model on the test sets of the 12 converter valves, as shown in Figure 12. We can see that removing any module from the proposed TransFNN model led to an increase in the error rate of overtemperature prediction, and the proposed TransFNN model improved the prediction accuracy by 6.58% compared with the original Transformer model.

### 5.5. Comparative Analysis of TransFNN: Assessing the Performance against Other Time-Series Prediction Algorithms

The Transformer model has demonstrated excellent performance in time-series prediction, but there are other traditional machine learning algorithms and deep learning models that offer certain advantages in this task, such as faster computing speeds or higher prediction accuracy. Compared to traditional time-series prediction models, such as random forest (RF), RNN, LSTM, and GRU, the Transformer model has two main advantages. Firstly, it uses a unique self-attention mechanism that balances the importance of different time positions in the sequence, thereby better capturing long-term dependencies in the data. This is particularly useful in time-series forecasting, as past values strongly influence future values. Secondly, the Transformer model is fully parallelizable, enabling it to process all elements of an input sequence simultaneously, making it much faster than a model that processes data sequentially.

Based on these advantages, we used the Transformer model as the module for implementing time-series prediction in our proposed TransFNN model. To evaluate the performance of TransFNN, we compared it with other time-series models, including RF, RNN, LSTM, and GRU, which represent a comprehensive range of hybrid models.

We conducted the experiment with the same dataset and experimental setup as in the previous section. For the evaluation index of RMSE, we calculated the average RMSE of the cooling water temperatures of all converter valves in the prediction test set for each model. For the evaluation index of the prediction time, we calculated the average time taken by each model to predict the valves’ cooling water temperatures at a single time point. We then calculated the RMSE reduction for each model compared to RF and the prediction time reduction for each model compared to the RNN, as shown in Table 3. We added the percentage of RMSE reduction and the percentage of prediction time reduction for the different models to obtain the total optimization percentage for these models. It should be noted that the weight given to RMSE reduction and prediction time reduction in evaluating time-series prediction models can depend on the specific application and requirements. In some cases, such as in real-time systems or applications in which speed is critical, the prediction time may be more important than the RMSE. In other cases, such as in scientific research or financial forecasting, the RMSE may be more important than the prediction time.

Table 3 summarizes the performance of the different time-series prediction models. As shown, when the weights given to RMSE reduction and prediction time reduction were the same, the Transformer model outperformed the other models in our proposed framework for predicting the cooling water temperature of a converter valve. However, since the real-time prediction of the overtemperature state of the converter valve was not required in this scenario, we gave a greater weight to RMSE reduction. This further decreased the overall performance ranking of the prediction model that used RF. Nevertheless, the Transformer model remained the best-performing module in this hybrid model. Therefore, the comprehensive performance of the TransFNN model proposed in this paper achieved satisfactory results.

## 6. Conclusions

This study proposed a novel approach to predicting the overtemperature state of converter valves, which was achieved by implementing the TransFNN model. The model combined an F-NN module, which coordinated the FCM algorithm and an ANN, with an improved Transformer model. This approach overcame the issue of low prediction accuracy in the original Transformer model when processing the cooling water temperature of a converter valve. The TransFNN model was able to accurately predict the overtemperature state of converter valves, which is essential for the effective maintenance and operation of converter stations.

The experimental results showed that the proposed TransFNN model outperformed four other models in terms of prediction accuracy. By removing some modules from TransFNN and applying it to predict the overtemperature state of 12 converter valves, the accuracy was 91.81%, which was 6.85% higher than that of the original Transformer model. The improved accuracy of the TransFNN model provides valuable information to the operation and maintenance personnel of converter stations, enabling them to punctually and appropriately adjust power dispatching and cooling measures, thereby avoiding equipment damage and resource waste. Additionally, the study enhanced the positional encoding method in the Transformer, making it suitable for predicting time series with strong periodicity in different time dimensions. This study also provided a new framework for predicting the time series of multi-source data, with data clustering as one of the pre-processing steps for improving the similarity of the data in the same operating conditions. This resulted in a deep learning model with improved learning abilities, leading to an overall improvement in prediction accuracy.

## Figures and Tables

**Figure 1 sensors-23-04110-f001:**
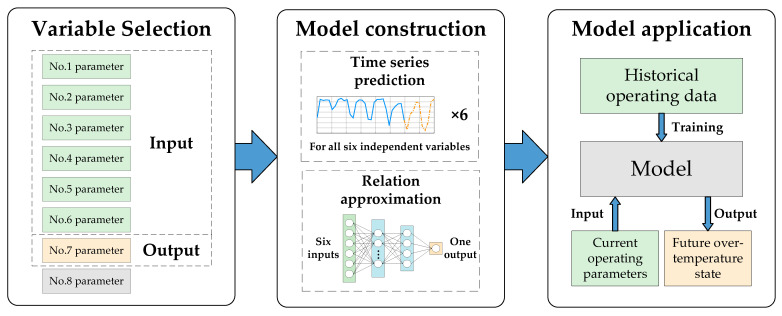
Flowchart of the overall scheme.

**Figure 2 sensors-23-04110-f002:**
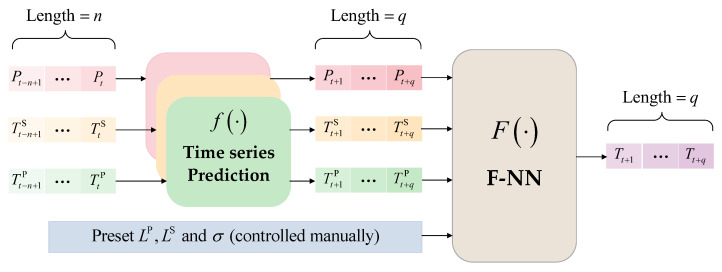
Schematic diagram of the prediction of a valve’s cooling water temperature.

**Figure 3 sensors-23-04110-f003:**
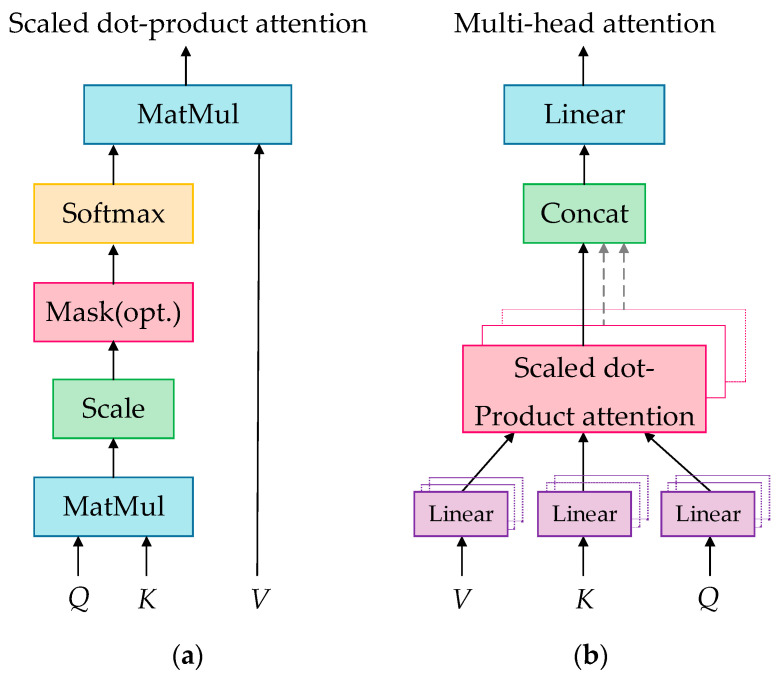
Two attention models: (**a**) scaled dot-product attention model; (**b**) multi-head attention model.

**Figure 4 sensors-23-04110-f004:**
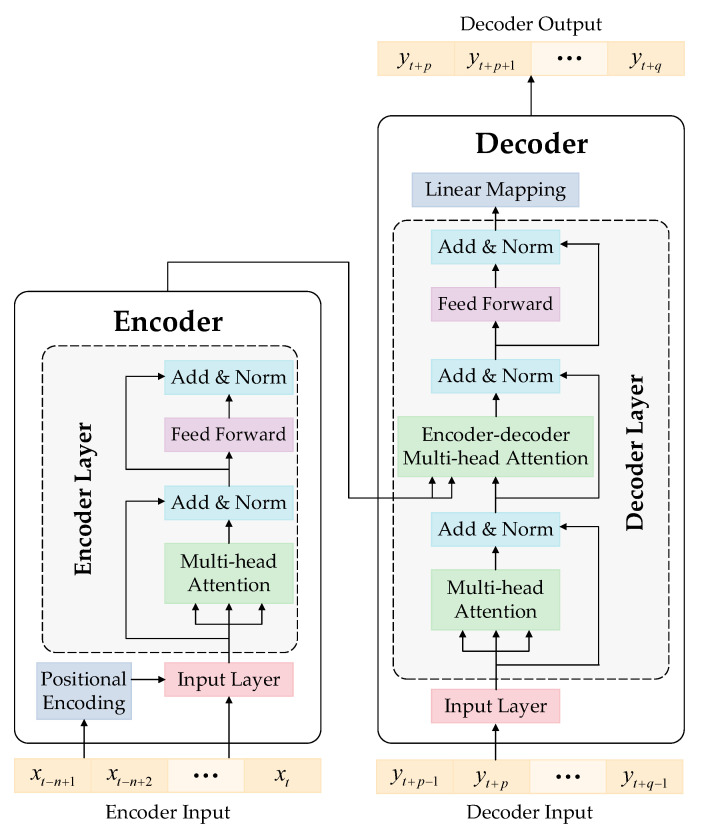
The Transformer model’s architecture.

**Figure 5 sensors-23-04110-f005:**
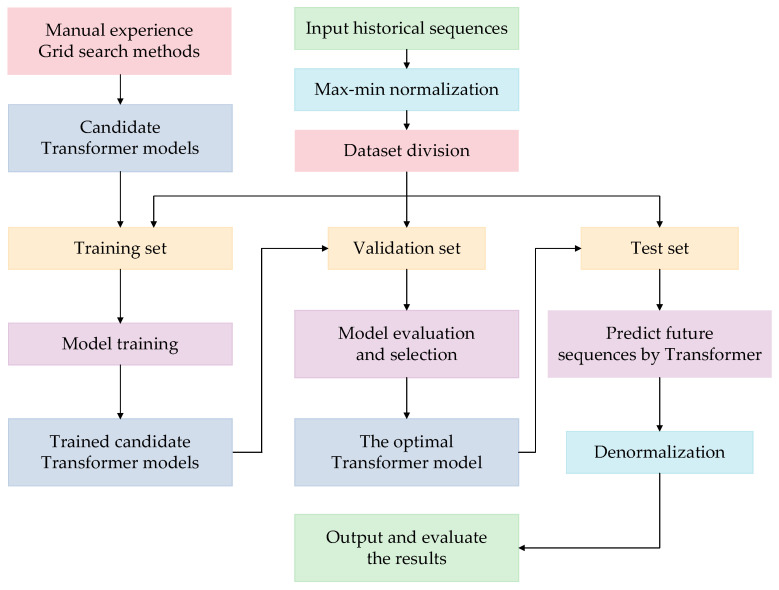
Flowchart of Transformer-based prediction of independent operation variables.

**Figure 6 sensors-23-04110-f006:**
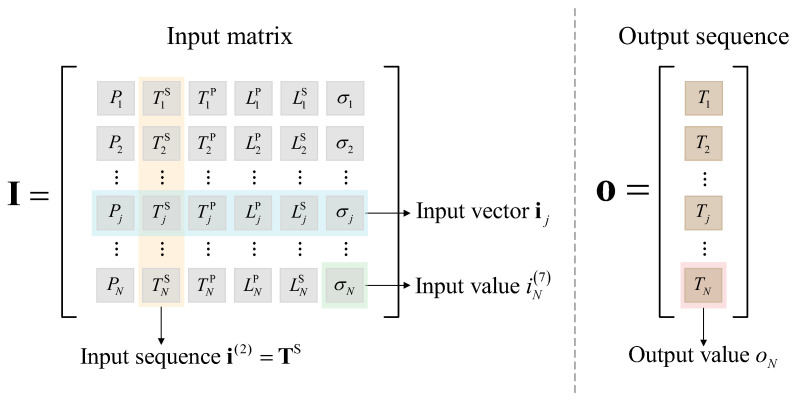
Original operating database of the converter valve cooling system.

**Figure 7 sensors-23-04110-f007:**
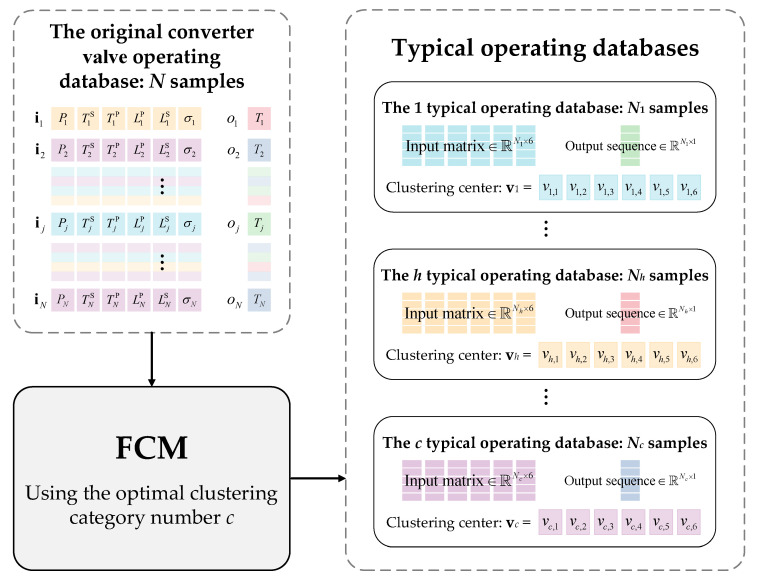
Diagram of the construction of a typical database for a converter valve.

**Figure 8 sensors-23-04110-f008:**
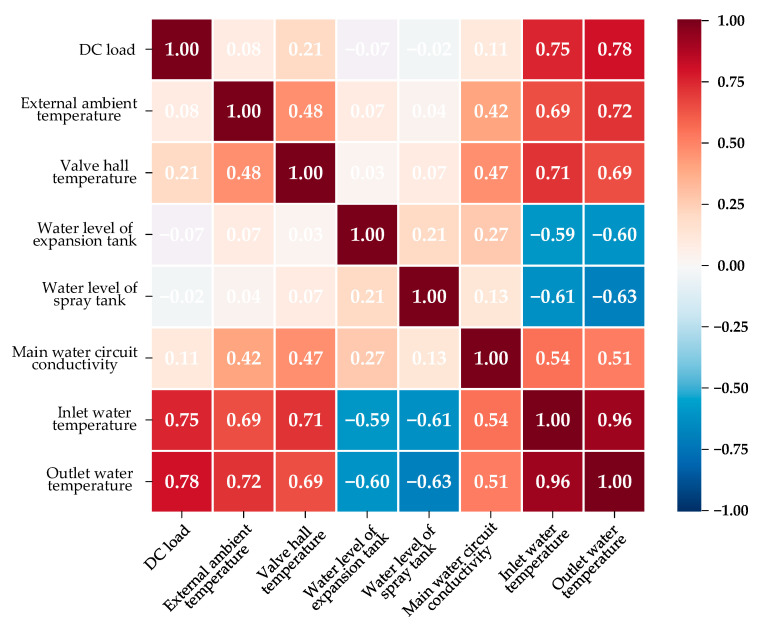
Pearson’s correlation coefficient analysis of eight operating parameters.

**Figure 9 sensors-23-04110-f009:**
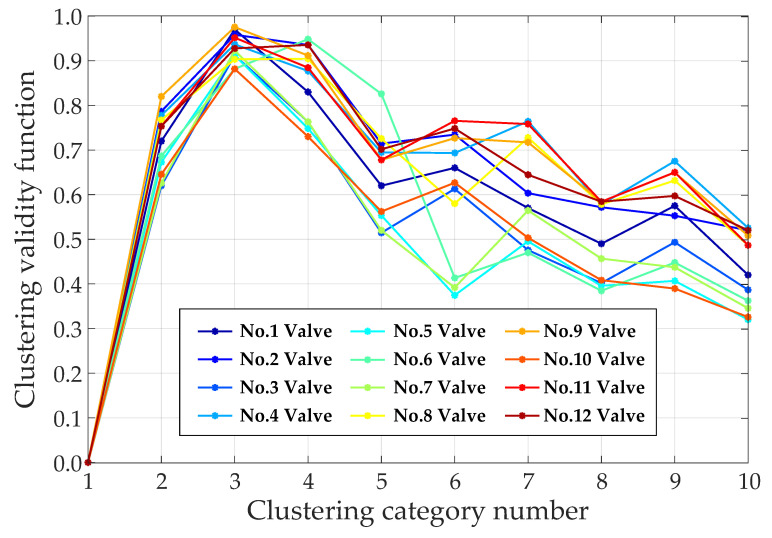
Clustering validity for different category numbers.

**Figure 10 sensors-23-04110-f010:**
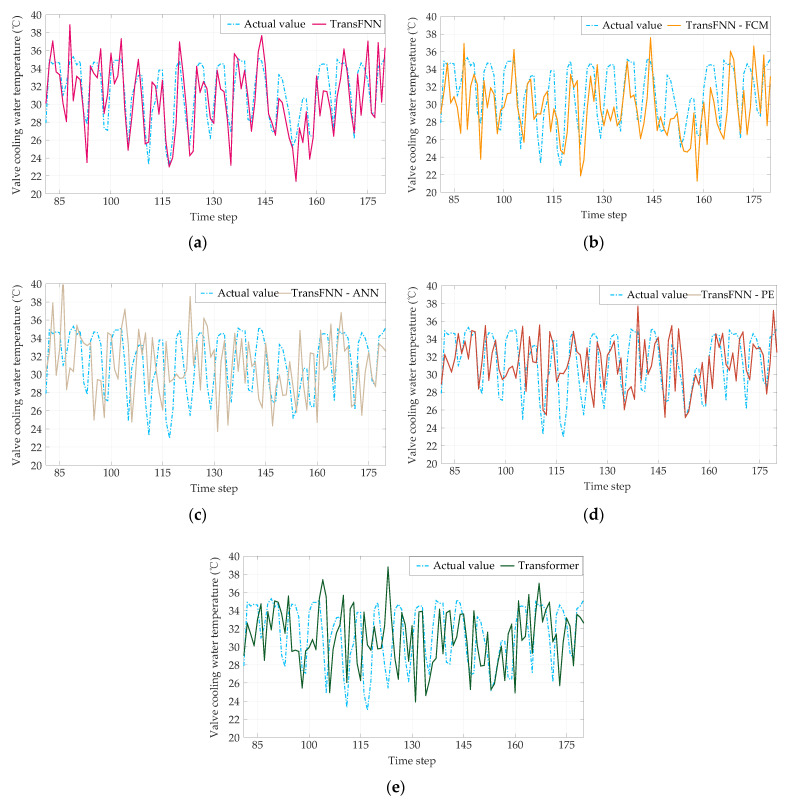
Prediction results of the five models: (**a**) TransFNN; (**b**) TransFNN−FCM; (**c**) TransFNN−ANN; (**d**) TransFNN−PE; (**e**) Transformer only.

**Figure 11 sensors-23-04110-f011:**
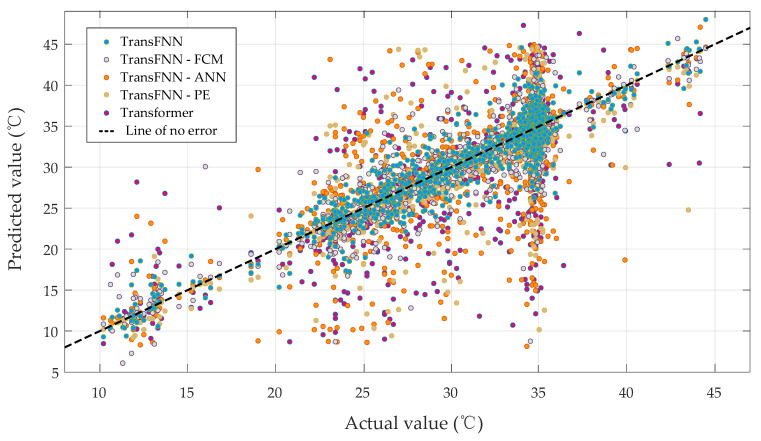
Scatterplot of the actual and predicted values.

**Figure 12 sensors-23-04110-f012:**
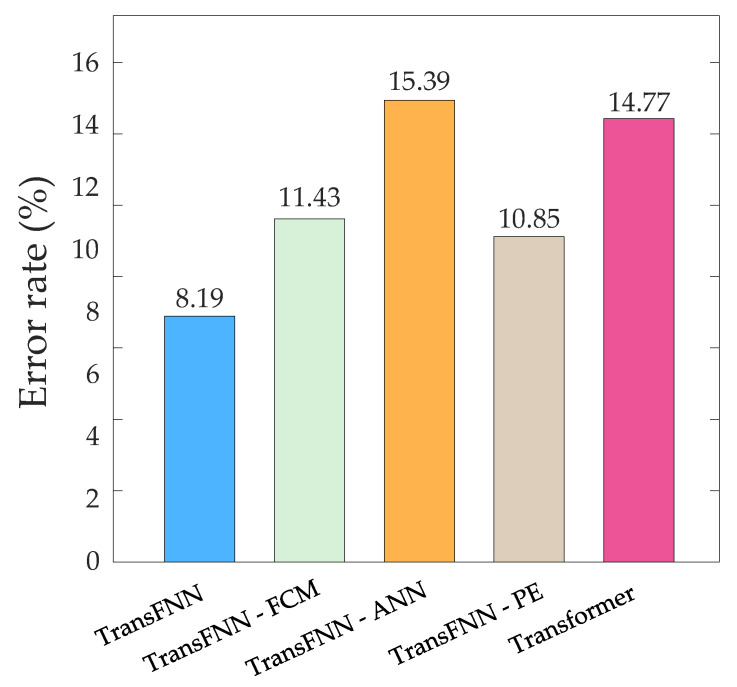
Bar chart of the error rates of the five models.

**Table 1 sensors-23-04110-t001:** Operating parameters of the cooling system of an HVDC converter valve.

Data Source Type	Parameter Name	Symbol	Variable Type
Electrical data	DC load (MW)	P	Independent
Environmental data	External ambient temperature (°C)	$T^{S}$	
	Valve hall temperature (°C)	$T^{P}$	
Water cooling system data	Water level of expansion tank (%)	$L^{P}$	
	Water level of spray tank (%)	$L^{S}$	
	Main water circuit conductivity (μs/cm)	σ	
	Valve cooling water temperature (°C)	T	Dependent

**Table 2 sensors-23-04110-t002:** Performance comparison of the five models.

Valve Number	Model	Performance Measures Obtained from Test Sets				Ranking
		$RMSE\left({}^{\circ}C\right)$	MAE(°C)	$sMAPE\left(\%\right)$	$R^{2}(\%)$	
No. 1	TransFNN	**2.55**	**2.13**	**8.11**	**92.45**	**1**
	TransFNN−FCM	3.68	3.22	11.75	87.66	3
	TransFNN−ANN	4.97	4.11	15.77	75.29	5
	TransFNN−PE	3.18	2.43	10.04	91.11	2
	Transformer	4.64	3.88	14.82	82.74	4
…						
No. 7	TransFNN	**2.13**	**1.97**	**7.92**	**94.86**	**1**
	TransFNN−FCM	2.94	2.54	10.33	90.83	2
	TransFNN−ANN	4.75	4.09	15.27	75.98	5
	TransFNN−PE	3.96	3.32	11.65	87.92	3
	Transformer	4.57	3.76	14.72	82.41	4
…						
No. 12	TransFNN	**2.69**	**2.07**	**8.48**	**92.15**	**1**
	TransFNN−FCM	3.98	3.28	11.63	88.23	3
	TransFNN−ANN	4.66	4.09	14.98	79.37	5
	TransFNN−PE	3.75	3.07	10.42	89.86	2
	Transformer	4.36	3.94	15.37	81.52	4

Note: Bold values denote the best performance measures among the models. A model with bold denotes the best model for each group. ‘A − B’ means removing module B from model A.

**Table 3 sensors-23-04110-t003:** Performance when using different time-series prediction models.

Model	Average RMSE		Average Prediction Time		Total Reduction	
	Value (°C)	Reduction (%)	Value (ms)	Reduction (%)	Value (%)	Ranking
RF	5.37	0	9.31	66.41	66.41	3
RNN	3.29	38.73	27.72	0	38.73	5
LSTM	3.07	42.83	21.96	20.78	63.61	4
GRU	2.88	46.37	19.75	28.75	75.12	2
**Transformer**	**2.52**	**53.07**	**13.21**	**53.34**	**106.41**	**1**

Note: For RMSE reduction, the benchmark is ‘’RF’’, and for calculation time reduction, the benchmark is ‘’RNN’’. The value of total reduction is the sum of the RMSE reduction and calculation time reduction. Bold values denote the best performance among the models.

## Data Availability

Not applicable.

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
