# Peer review of "TransFNN: A Novel Overtemperature Prediction Method for HVDC Converter Valves Based on an Improved Transformer and the F-NN Algorithm"

_sensors, 2023, doi:10.3390/s23084110_

Round 1

Reviewer 1 Report

This manuscript advocates the utilization of a transformer model to predict overtemperature cases of HVDC converter valves. Feedback on this manuscript is as follows:

1. The experimental design is inadequate. The authors concede that numerous models exist that can operate with time series data, yet they have opted solely for a transformer model. It appears that the selection was based on hype rather than a rigorous selection of baselines. Implementing a more stringent set of baselines is necessary. Furthermore, it is imperative to consider that transformer models require more energy and hardware resources. Thus, the possibility of a simpler model such as a random forest model being viable must be addressed.

2. The figures included in the manuscript are illegible, and therefore, the resolution of the figures must be enhanced.

Reviewer 2 Report

This paper proposes a novel approach to predicting the valve overtemperature state. However, there are still some questions to be addressed.

1.The quality of the pictures in the paper needs to be improved.

2.Is the data typical, and is the data set large enough for training?

3.The conclusion needs to be more concise.

4. It is suggested to use other time series models for comparison, such as the LSTM model mentioned in this paper.

Reviewer 3 Report

This paper provides a novel approach (Transformer-FCM-NN) to predicting the valve over temperature state, acting as a data-driven tool for the operation and maintenance personnel to adjust the valve cooling measures timely, effectively and economically. The results of the quantitative experiments showed that the proposed TransFNN model outperformed the comparative models. The manuscript is not ready for publication and requires a revision. The specific comments are listed as below:

1.      Why the speed of internal water circuit be concerned as a factor influencing the temperature of valve’s cooling system in Mathematical model for valve cooling water temperature predictions?

2.      The description of the neural network used in this paper is too lengthy.

3.      According to the description in Figure 9 in this paper, perhaps it should be written no.1 to no.12 on the 496th line.

4.      All pictures in this paper are not clear.

5.      Incorrect serial number on the 398th line.

Reviewer 4 Report

First of all, thank you very much for choosing our journal for your article. It was indeed a very comprehensive and successful work. If the corrections I mentioned are made, the article is suitable for publication for me.

- Please redesign the figures throughout the entire article. The image quality of the figures is very poor. It makes the reader very difficult while reading.

- In line 210, after process, when should be written in lowercase

- I don't understand what is written here

- In line 405, the written twice please delete one

- In line 538, water must be written in lowercase

Round 2

Reviewer 1 Report

Thanks for addressing my feedback

Reviewer 2 Report

Accept in present form.